# Lockdown-Associated Hunger May Be Affecting Breastfeeding: Findings from a Large SMS Survey in South Africa

**DOI:** 10.3390/ijerph19010351

**Published:** 2021-12-30

**Authors:** Nazeeia Sayed, Ronelle Burger, Abigail Harper, Elizabeth Catherina Swart

**Affiliations:** 1School of Public Health, University of the Western Cape, Bellville, Cape Town 7535, South Africa; 2Department of Economics, University of Stellenbosch, Stellenbosch 7602, South Africa; rburger@sun.ac.za; 3School of Public Health, University of Witwatersrand, Parktown, Johannesburg 2193, South Africa; abigailjroso@gmail.com; 4Department of Dietetics and Nutrition, University of the Western Cape, Bellville, Cape Town 7535, South Africa; rswart@uwc.ac.za

**Keywords:** breastfeeding, hunger, food security

## Abstract

The impact that the COVID-19 pandemic has had, and will continue to have, on food security and child health is especially concerning. A rapid, Short Message Service (SMS) Maternal and Child Health survey was conducted in South Africa in June 2020 (*n* = 3140), with a follow-up in July 2020 (*n* = 2287). This was a national cross-sectional survey conducted among pregnant women and mothers registered with the MomConnect mhealth platform. Logistic regression was conducted to explore the associations between breastfeeding, maternal depressive symptoms, and hunger in the household. High breastfeeding initiation rates and the early introduction of other foods or mixed milk feeding were found. The prevalence of depressive symptoms in this survey sample was 26.95%, but there was no association between breastfeeding behaviour and depressive symptom scores (OR = 0.89; 95% CI: 0.63, 1.27). A positive correlation was found between not breastfeeding and not going to the health clinic. The odds of hungry mothers breastfeeding were significantly lower (OR = 0.66; *p* = 0.045). This result also holds in a multivariate framework, including covariates such as depressive symptoms, attendance of a PHC facility, and whether the infant was older than 3 months. Support for breastfeeding must include support, such as economic support, for breastfeeding mothers, to enable them to access nutritious diets. Mothers also need reassurance on the quality of their breastmilk and their ability to breastfeed and should be encouraged to continue to attend the health clinic regularly.

## 1. Introduction

Improvement of breastfeeding rates remains a key strategy globally to ensure food security and optimal development for the first 6 months of life [1,2]. There has been some progress reported on increased exclusive breastfeeding rates in South Africa, but rates still remain low [3]. Several factors adversely impact mothers’ breastfeeding ability and breastfeeding behaviour. Food insecurity, which has declined steadily in South Africa (2002 to 2017) [4], is one of these factors. In response to the COVID-19 pandemic, South Africa imposed a strict national lockdown on 27 March 2020, which resulted in severe job and income losses, as well as food insecurity and mental health concerns [5]. This study was inspired by this context and the literature on lower breastfeeding rates and earlier cessation of exclusive breastfeeding seen with increased severity of food insecurity experienced by the mother and the household [6,7,8,9].

The data used for this assessment were derived from a Short Message Service (SMS) Maternal and Child Health (MATCH) survey, implemented in June–July 2020 as a rapid assessment of the impact of the COVID-19 pandemic in South Africa on the recently improved breastfeeding rates. This was a national, cross-sectional survey conducted among pregnant women and mothers registered with the MomConnect mhealth platform in South Africa. The purpose of the survey was to assess nutrition (hunger/food security, breastfeeding), depressive symptoms, access to antenatal care, clinic visits, infant vaccinations, and anti-retroviral therapy, as well as their impediments to access. This paper reports on findings from the survey related to breastfeeding, and its relationship with hunger, depressive symptoms, and clinic visits.

## 2. Methods

### 2.1. Study Design

Data were generated from a national, cross-sectional survey, using mobile SMS for recruitment and data collection from pregnant women and mothers in the public health sector in South Africa using the MomConnect mhealth platform, which is estimated to have enrolled more than half of all women attending the public sector antenatal care services [10].

The enrolled women received an invitation to join the SMS survey. They could respond with ‘JOIN’ to participate, ‘STOP’ to not participate, or ‘MORE’ if they needed further information. There was no cost for participating in the survey. Those who participated in the survey received R10 (10 South African Rands; 0.59 US dollars) in airtime as a token of appreciation.

A response rate of 21% (*n* = 3140) was achieved for the first survey. Respondents were invited on the afternoon of 24 June 2020, and the survey closed on 30 June 2020. Of these respondents, 41.5% were post-birth, and the rest were pre-birth individuals. A follow-up survey invitation with further questions was sent on 2 July, and the second survey ended on 5 July 2020. Of the 3140 individuals that responded to the first survey, 2287 (72.8%) also responded to the follow-up survey. Of these respondents, 40.7% were post-birth, and the remainder were pre-birth individuals.

A targeted post-birth sample of 1000 would have enabled detection of a 10.3% minimum effect size, assuming a mean breastfeeding rate of 67%, based on the South African Demographic and Health Survey 2016 [11]. The achieved sample for our breastfeeding variable was 985, but due to the small deviation from the target sample, the minimum detectable effect size remained at 10.3%.

### 2.2. Permission and Ethics Approval

Permission for the survey was obtained from the National Department of Health, and ethics approval was obtained from the University of Stellenbosch’s Research Ethics Committee for Social, Behavioural, and Education Research (Project 14926 on 15 June 2020).

### 2.3. Sampling

A self-weighted sample of 15,000 pregnant women and mothers with children under 12 months was drawn from the database of MomConnect users. The sample was stratified based on province, gestational age (pre-birth) or age (post-birth) of their baby and their type of phone (i.e., smartphone or a basic phone, as a proxy of income) to achieve a representative sample of MomConnect users. Further sample information is available in the Appendix A provided (Appendix A).

### 2.4. Data Collection Instruments and Variable Definitions

Only the instruments pertaining to the variables analysed in this paper are described here. Due to the short format of the SMS survey, and the need for simple questions in English that could be understood by the mothers, data collection tools had to be modified.

Depressive symptoms were assessed using a modified version of the PHQ-2 tool [12]. This Likert-like scale asked two questions about the prevalence of depressive symptoms in the past week: ‘In the last 7 days, have you felt hopeless, down, or depressed?’ and ‘In the last 7 days, have you felt little interest or pleasure in doing things?’ Respondents then proceeded to indicate the regularity with which they experienced negative feelings: ‘no’; ‘yes, a few days’; ‘yes, most days’. Respondents were then assigned a continuous score between 0 and 6 with each increased unit indicating increased severity of symptoms. The literature used a cut-off of either 2 or 3. We used a cut-off of 2 to indicate the likelihood of depressive symptoms but confirmed the robustness of all results using a cut-off of 3.

Questions on hunger in the first survey enquired if a child in the household had experienced hunger in the last seven nights and whether an adult in the household had experienced hunger in the last seven nights. In the follow-up survey, the hunger questions focused on whether the respondent had experienced hunger in the last seven nights and a child had experienced hunger in the past seven nights. We used the guidelines for interpretation of the household hunger scale by Ballard et al. (2011) [13] and scaled it accordingly for a reference period of 7 days. Thus, a hunger frequency of once in 7 days was classified as ‘rarely’; 2 times in 7 days was classified as ‘sometimes’ (equivalent to 3–10 times in 30 days); 3 or more times in 7 days was classified as ‘often’ (equivalent to >10 times in 30 days). To investigate the relationship between breastfeeding and hunger, ‘rarely’ was recategorised as not hungry/food secure after combining cases with a ‘never’ response/at risk of food insecurity; ‘sometimes’ and ‘often’ were classified as hungry/food insecure.

Breastfeeding information was collected only in the follow-up survey (July 2020). Currently breastfeeding was determined by a positive response to the question ‘Yesterday, did you breastfeed your baby?’. The next question asked ‘Yesterday, did you feed your baby formula or porridge such as Nestum?’ to assess mixed feeding. If the answer to the first question was yes, and no to the second question, then a follow-up question was posed to identify predominant breastfeeding by asking if only breastmilk was given, if only formula or cereal was given, or if both breastmilk and formula or cereal was given in the seven days prior. These three questions were combined to create an indicator for predominant breastfeeding, categorising women as predominant breastfeeding if they breastfed their baby the previous day but did not feed them formula or porridge and also confirmed that over the past seven days they only gave their baby breastmilk (and no formula or infant cereal).

Primary health care (PHC) facility attendance of pregnant women and mothers was assessed in the first survey by asking them when they last attended the clinic. To capture clinic attendance, we created a binary variable that was 1 if respondents had visited a clinic in the previous 2 months, and 0 if they had not.

The follow-up survey included a question asking women to share their main concerns. It was an open-ended question and limited only by the 160-character limit of the SMS.

Table 1 outlines the data collected in the first and follow-up rounds of the national survey. Data on child hunger were the only data collected in both rounds of the survey. Depressive symptom data were only collected in the first round of the survey, and all other data used in this study were collected in the follow-up round of the survey. The questions from the 2 surveys are available in the Appendix A provided (Appendix A).

### 2.5. Statistical Analysis

The two waves of the survey were analysed as a panel due to the proximity in time. Due to the survey format, the researchers were limited in what questions they could include, which in turn has constrained the study’s ability to understand the relationships and patterns in more depth. Univariate analysis examined the prevalence of breastfeeding practices and hunger in the household. Bivariate analysis was employed to explore the associations between breastfeeding and maternal depression, and breastfeeding and hunger, as well as between breastfeeding and clinic attendance. In all cases a Pearson’s chi-squared test was used to assess the significance of the relationship. We also included multivariate analysis where we explored whether the relationship between maternal hunger and breastfeeding was robust to the inclusion of PHC facility, depressive symptoms of the mother and the baby’s age in a multivariate logistic regression. Throughout the paper, we employed a 5% cut-off for assessing the significance of a statistical relationship. Stata version 15.2 (Stata Corp, College Station, TX, USA) was used for the statistical analysis.

## 3. Results

The respondent’s responses to the question about what worried them most at the moment provided some confirmation of our concern that the pandemic had affected hunger, nutrition, and breastfeeding. Typical responses included the following: *‘I worry about losing my life or my kids due to COVID-19 and not having food in the house, because as a breastfeeding mom I have to eat so that I can produce milk’*; *‘I am concerned about going to bed hungry when I have to breastfeed’*; *‘I am worried about my health and wellbeing together with good nutrition since I am breastfeeding.’*

### 3.1. Breastfeeding Practices

We found that amongst mothers with infants of 3 months or younger, 94.1% said that they breastfed the previous day and 72.2% were breastfeeding predominantly and not using formula milk or other foods. Of the sample of infants who were 5–6 months old, 93.1% of infants were still receiving breastmilk, but only 28.6% were breastfed predominantly.

### 3.2. Breastfeeding and Maternal Depressive Symptoms: Breastfeeding Behaviour Was Not Associated with Depressive Scores

The prevalence of depressive symptoms in this survey sample was 26.9%. There was not a significant association between breastfeeding behaviour and depressive symptom scores (OR = 0.89; 95% CI: 0.63, 1.27).

### 3.3. Breastfeeding and Clinic Attendance: Breastfeeding Mothers Were More Likely to Not Miss Their Primary Health Care Facility Visits

This survey found that women who had not visited a PHC facility recently (in the past two months) were significantly less likely to breastfeed (Table 2). Overall, 85% of mothers who had been to a PHC facility recently breastfed their baby currently, while 75% of those who had not been to the clinic breastfed their baby currently. A test of proportions confirms that the difference between these means was significant (*p* = 0.004). There was also a significant relationship between predominant breastfeeding and recent PHC facility attendance. Predominant breastfeeding was practiced by 17% of mothers who had been to the clinic recently and by 11% of those who had not been to the clinic recently (*p* = 0.016).

### 3.4. Breastfeeding and Hunger: Hungry Mothers Were Less Likely to Breastfeed (Compared with Mothers Who Did Not Report Hunger)

Approximately, 18.3% of respondents reported going to bed hungry over the past 7 days. Of the respondents who went to bed hungry, 28.2% rarely went to bed hungry, 37.9% sometimes went to bed hungry, and 28.0% often went to bed hungry. Overall, 1 in 20 (5.9%) respondents who went to bed hungry did not want to provide information on how often they went to bed hungry or said they did not know.

Children going to bed hungry (asked in the second wave of the survey) was not associated with a decreased prevalence of breastfeeding (captured in the same round of the survey). However, mothers who reported that they went to bed hungry in the past seven nights were significantly less likely to report breastfeeding in the previous day (OR = 0.66; *p* = 0.045). Table 3 shows that this result also holds in a multivariate framework, including covariates such as depressive symptoms, attendance of PHC facility, and whether the infant was older than 3 months. The multivariate logistic regression’s odds ratios for the covariates (depressive symptoms and attendance of PHC facility) are aligned with the significance and the direction of association reported for the bivariate relationships.

## 4. Discussion

### 4.1. Breastfeeding Practices

The findings on breastfeeding practice found in this survey resonate with previous findings in South Africa. Breastfeeding initiation rates in South Africa are high (ranging from 75 to 100%), but continued breastfeeding varies, and there is widespread early introduction of foods and liquids other than breastmilk/formula milk [14]. The South African Demographic and Health Survey 2016 reported that 32% of infants younger than six months were exclusively breastfed [11].

### 4.2. Breastfeeding and Maternal Depressive Symptoms

Although this study found no association between breastfeeding and depressive symptoms, depressive symptoms could influence a mother’s breastfeeding self-efficacy [13] and may result in a shorter duration of breastfeeding or a greater likelihood of breastfeeding cessation [15,16]. The bidirectional relationship between depressive symptoms and breastfeeding may be due to breastfeeding, leading to better mother–infant interaction, the release of oxytocin, and decline in cortisol levels, which may improve maternal mood and reduce feelings of stress [17].

### 4.3. Breastfeeding and Clinic Attendance

This relationship between clinic visits and breastfeeding found in this study could be attributable to the role of primary health care facilities in supporting and encouraging mothers to breastfeed. Alternatively, it could reflect an endogenous (intraperson) relationship between caregiving and care-seeking behaviours, reflective of intergenerational transmission of attachment [18]. It is argued that attachment, one of many behavioural systems, reflects intergenerational caring practices. The documented replacement of mother–child caring by grandmother–child caring in the South African context [19] may affect the caregiving (breastfeeding) and care-seeking (PHC visits) behaviours of mothers. It is also possible that not breastfeeding and not visiting clinics are related to other factors not assessed in this study.

More regular clinic attendance may allow for an increase in knowledge on breastfeeding, improved self-efficacy for breastfeeding, and emotional support to breastfeeding [20]. A meta-analysis found that breastfeeding education and support were associated with both breastfeeding initiation and continuation [21]. There are some indications that postnatal support may be slightly more effective than antenatal breastfeeding education [22], but both may be required to impact breastfeeding continuation [23]. Ultimately, breastfeeding support in a combination of settings (health services, the workplace, the home family, and community), along with appropriate policy, is required for improved breastfeeding [24].

### 4.4. Breastfeeding and Hunger

Household hunger levels in South Africa have been decreasing; the 2017 estimates reveal that one in five households are food insecure and experience hunger [4]. Hunger in pre-birth and post-birth women specifically has not been assessed nationally before in South Africa, so the findings of this study on hunger cannot be compared with pre-COVID estimates.

The current study does not allow conclusions beyond associations between breastfeeding and hunger within the COVID-19 context. Previous studies have identified a link between food insecurity or hunger and breastfeeding. A Canadian study observed that more than half of food-insecure mothers had ceased to breastfeed exclusively by 2 months [7]. A longitudinal cohort study in Kenya [25] found that maternal hunger was associated with lower rates of breastfeeding, but that mothers with greater self-efficacy across all levels of hunger were more likely to exclusively breastfeed than those with poor self-efficacy. Hunger in mothers may contribute to a perception of milk insufficiency and undermine confidence in their breastfeeding ability. A study in Kenya [26] found that there were greater odds that a woman in a food-insecure household would consider her breast milk insufficient and that they would not breastfeed for 6 months. For every one-point increase in the household food insecurity score, another Kenyan study [27] found that there was a decrease in breastmilk intake by the infant. A 2015 Canadian study postulated that household food insecurity could be a predictor of breastfeeding initiation due to concerns on the cost of alternate feeding, but concerns over their own food and nutrient intake and the quality or quantity of breastmilk they produced may lead some mothers to introduce formula milk [28]. Reducing hunger in mothers by improving household food security and improving breastfeeding self-efficacy in mothers could lead to higher rates of exclusive breastfeeding [25,29].

COVID-19 is likely to increase stress levels in women. The stress that mothers from low-income households in South Africa experience could include concerns about their breastmilk supply, their access to food, having regular meals, and relationship difficulties in the home [29]. Women, in particular, face an increased care burden due to COVID-19, and this further increases the stress on them [30].

### 4.5. Limitations

A key limitation of the findings presented here is the cross-sectional nature of this survey. Hunger and depressive symptoms have various dimensions that may change with time. The questions and tools used to assess these measures may also affect the results. The survey did not assess exclusive breastfeeding, as water and consumption of other substances commonly given to infants were not assessed. The findings of this survey cannot be generalised to the South African population, as only mothers who subscribed to the MomConnect platform and who chose to participate constituted the study sample. Mothers who accessed private health care services were thus excluded from this study. Although there was no significant difference in the response rates of the women based on the socioeconomic quintile of the primary care facility where they registered for MomConnect, there may have been systematic bias in the response based on information not available in the dataset. Due to the reliance on SMS communication, the survey could not gather information on a large range of determinants of breastfeeding.

### 4.6. Recommendations to Support Breastfeeding

Based on the findings of this survey that mothers experiencing hunger are less likely to breastfeed, the authors recommend a paradigm shift in efforts to improve breastfeeding, to be more comprehensive. This includes strengthening of enabling capabilities [31] such as health-seeking behaviour and self-efficacy; direct nutrition support of pregnant and breastfeeding women; inclusion of the value of breastfeeding beyond only an immediate superior nutritional and hygienic feeding method to include the food security and aspects related to noncommunicable disease prevention; the inclusion of a whole-of-society approach [32].

Breastfeeding needs to be protected, promoted, and supported more than ever, as this can impact the health of future generations and also offers all infants a fairer/more equal start in life. In addition, mothers need to be encouraged to continue to attend the health clinic. Maternal and child nutrition needs to be prioritised alongside the COVID-19 response, or we will see the devastating impact of malnutrition, such as stunted growth, diminished cognitive potential, and an increased burden of noncommunicable diseases in the coming generations [33].

Among the four lifesaving interventions to be prioritised during COVID-19, the UN recommends mass communication to caregivers and families to protect, promote, and support breastfeeding in children aged 0–23 months [34]. Further communication, supporting the breastfeeding mother to enable her to breastfeed and feel competent to do so is a critical gap that needs to be addressed. Lee (2016) [35] argues that the ethical obligation to breastfeed a child extends beyond the mother to the broader society which has to provide social and economic support to enable breastfeeding, and this may also extend to providing food to the hungry breastfeeding mother as a priority (or enabling her to access nutritious food). Mothers also need reassurance on the quality of their breastmilk and their ability to breastfeed. From a health perspective, the benefits of breastmilk are undisputed and contrary to hungry mothers who cease breastfeeding in the belief that this will have a better outcome for their infants.

The effect that COVID-19 has had on breastfeeding advice and support in South Africa is not known. South Africa’s history and learnings from HIV and infant feeding guidance should not be forgotten, and efforts need to include reaffirming the benefits of breastfeeding among our health care workers. Evidence suggests that health care workers’ infant feeding advice is undermined by implementing national guidelines within local contexts and fears of HIV transmission [36]. Aside from the nutritional superiority, the safety and immune benefits of breastmilk may need to be asserted to counter concerns during the continuing COVID-19 pandemic.

Support for breastfeeding cannot be distanced from support for the mother. Social protection policies such as the child support grant top-up and the COVID-19 social relief of distress grant in South Africa can assist women to reduce hunger while breastfeeding. In addition, mothers need reassurance about the quality of their breastmilk and their ability to breastfeed. Health care professionals and community health workers are trusted experts in this context and should be empowered to support breastfeeding mothers.

## 5. Conclusions

This study sought to investigate whether recently improved breastfeeding rates [3] were affected by the COVID-19 pandemic and the lockdown-associated changes in economic and social circumstances and to assess the association between breastfeeding, maternal depressive symptoms, maternal hunger, and clinic visits. No association was found between breastfeeding practice and maternal depressive symptoms in this study. The results show that hunger and not attending the clinic were associated with reduced breastfeeding. This confirms the importance of adequate nutrition, access to health care, and other structural barriers to breastfeeding uptake.

## Figures and Tables

**Table 1 ijerph-19-00351-t001:** Data collected in the first and follow-up round of the national survey.

	First Survey 24–30 June 2020 (*n* = 3140)	Follow-Up Survey 2–5 July 2020 (*n* = 2287)
Concerns about COVID-19	No	Yes
Currently breastfeeding	No	Yes
Mixed feeding	No	Yes
Predominant breastfeeding	No	Yes
Maternal depressive symptoms	Yes	No
Clinic attendance	Yes	No
Respondent (Mother) Hunger	No	Yes
Child Hunger	Yes	Yes
Household (Adult) Hunger	Yes	No

**Table 2 ijerph-19-00351-t002:** Cross-tabulation of breastfeeding and attendance of PHC facility.

Recently Attended PHC Facility	Currently Breastfeeding	Predominantly Breastfeeding
Yes	632 (85.0%)	126 (17.2%)
No	175 (74.8%)	25 (10.6%)

**Table 3 ijerph-19-00351-t003:** Logistic regression of correlates of breastfeeding *.

	Adjusted Odds Ratio (95% Confidence Interval)	*p*-Value
Maternal hunger	0.66 (0.44–0.99)	0.05
Depressive symptoms	0.93 (0.64–1.34)	0.69
Recent PHC facility attendance	1.67 (1.16–2.42)	0.01
Baby age 3–12 months	0.35 (0.18–0.69)	0.00
Constant	9.23 (4.36–19.57)	0.00
Observations	943	

* Note: The analysis used a logistic regression model to explore the relationship between breastfeeding and maternal hunger. Depressive symptoms, recent PHC facility attendance, and the baby’s age were included as covariates.

## Data Availability

The datasets analysed during the current study will be available in the DATAFIRST repository, [https://www.datafirst.uct.ac.za/dataportal/index.php/catalog/central/about] (accessed on 21 December 2021).

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
