# Peer review of "Lockdown-Associated Hunger May Be Affecting Breastfeeding: Findings from a Large SMS Survey in South Africa"

_ijerph, 2021, doi:10.3390/ijerph19010351_

Round 1
Reviewer 1 Report
Review of IJERPH-1407182-v1
This manuscript is much improved. At this point, given that the inherent problems of web-based survey research with unpredictable respondents can’t be fixed, it is as good to go as it will ever be (modulo some comments below). It is not clear that the data presented represent a lockdown-induced change, since no question in the survey addressed the reason for food insecurity or its timing. This could be addressed somewhat by using the “worry” question to see whether those who were worried about food were less likely to breastfeed in a multivariate model.
As a matter of reality, if a woman stopped breastfeeding when her child was 4 months old, and then lockdown came when the child was 5 months old, it is unlikely that she would be able to restart breastfeeding. This is relevant to the relationship between prevalence of breastfeeding and child age.
I stand by my general comments that paper titles, tables, and figures should indicate the geographical source(s) of the data and the calendar time they were collected. Tables should also indicate the method of analysis, if relevant.
To speed things up, I’m making English suggestions.
Specific comments:
Line comment
42 (?) “was” should be “were”
75 They are saying that 985 is practically the same as 1000, so the minimum detectable OR is the same. The way they say it is a bit awkward.
Please add “detectable” after “minimum.”
101-102 What is the basis for the cutoff they chose for depression (2)? It might be useful to do a sensitivity analysis with other cutoffs.
148 “Pearson’s” needs an apostrophe (as I have written it).
“multivariate” should be followed by “logistic.”
Table2 Is this table necessary?
194 “of” should be “at.”
Table 2 Why did they use robust standard errors? The outcome is only measured once and seems to be a good binary variable. If they have a good reason, they should explain it in the Methods section.
I assume that this is “any breastfeeding.” If it is “predominant breastfeeding,” please say so.
Please note the method of analysis.
Have the authors explored interactions among their variables? Given that this is basically an exploratory paper, adding a few more explorations won’t hurt.
Table 3 If they use the levels of hunger frequency, they can do a trend test, which might make the current result, with its marginal p-value, stronger.
208-210 this is speculation.
214 “mothers’” should be “mother’s.”
230 It is also possible that women who aren’t breastfeeding don’t want to go to PHC, because they don’t want to be lectured about breastfeeding.
321-322 “was” should be “were.”
The general discussion (multiple locations) of women’s perception that if they aren’t well nourished, they won’t be able to make good breastmilk. This may be a consequence of the type of education given (“eat well so you can breastfeed adequately”).
Reviewer 2 Report
Dear Authors,
This is a very significant study and its great to see that you have written on this aspect. I have some concerns with the methodology. Specifically:
Introduction
You mention barriers and factors that negatively impact mothers breastfeeding, what are those?
More background information is required of the women breast feeding and behaviour, to understand the lockdown effects. The method should remain in the methods section so delete this from here as it is repeated.
Methods
This section is not comprehensive. It will be great if you could add a descriptive information on the women, their life styles etc.
Table 2 – It will be good to add the significant rates and mention what analysis you did to achieve those results you mention in the text.
Discussion and Conclusion – Though the study concluded that the lock down had a negative impact on mothers breastfeeding due to hunger, there are many other covariates and confounding factors involved that are not mentioned here that could have an effect along with the lockdown. Information such as demographics, where they live, the family structure, etc could have an influence on their health and wellbeing and hence breast feeding.
More information and analysis is required.
Round 2
Reviewer 2 Report
Thanks for addressing the comments and fixing those.
Author Response
thank you - your feedback is appreciated
This manuscript is a resubmission of an earlier submission. The following is a list of the peer review reports and author responses from that submission.
Round 1
Reviewer 1 Report
This paper is on the relationship of breastfeeding and hunger in the lock-down period of COVID-19 in South Africa. The topic is interesting and the paper is written in clear English. However there are several flaws in the paper. 1. The format of the paper is not in accordance with the format required by this journal. Research methods, conclusions and discussions need to be written separately. 2. Breastfeeding can not be determined by a positive response to one question “Yesterday did you breastfeed your baby?”. 3. Although the sample was stratified based on province, gestational age or age of their baby and their type of phone. There is no detailed description and analysis of this factors.
Reviewer 2 Report
Review of IJERPH 1301893
I gather that this paper format is supposed to be much less formal than the usual one, but this paper does not give the reader enough information to know the extent to which the results relate to the covid-19 lockdown.
Line Comment
51 “self-weighted” I am unfamiliar with this term, but take it to mean that the sample was chosen to be proportionate to the whole set of women of MomConnect with regard to some set of characteristics considered relevant, which I guess are the variables they mention right after.
56 How does R10 compare to the cost of answering the survey? Did it just cover the costs, or was it a bit of a reward?
How did the respondents compare to the overall MomConnect population? Is there selection bias?
105 “By six months” This language is more appropriate for a cohort-type study where children are followed. This is a cross-section. Consider rephrasing to something like “Of the children who were 6 months old,…” (I assume that there was some sort of window around 6 months, or that you have integer months completed. Which?) And, of course, children who are over 6 months old should be getting food in addition to breast milk.
129-145 Perhaps this is normal for this journal, but this part looks a lot like the Discussion section.
154 What was the minimum detectable odds ratio with your sample size? (depends on prevalence of factor you are analyzing).
Also, please be clear about when you are reporting univariate results and when multivariate/adjusted results.
Table 1 Where did 978 come from? No need to report the value of chi-squared, but nothing before said you were going to use this method.
172-179 It is also possible that both not attending the PHC and not breastfeeding are related in a causal way to some third variable (or multiple variables), such as a poor economic circumstances or a change in economic circumstances.
180-188 More prose that sounds like the Discussion section.
196 “MumConnect” or “MomConnect” as above?
I would have liked to see a weighted multivariate analysis (in a table), to account for likely selection bias and for the influence of multiple factors. Although the authors say they used logistic regression, they present odds ratios only rarely. As it is, I don’t know what all the questions were or how they interrelate. While we all understand that covid-19 had caused huge economic dislocation, it is not clear from what is presented that either the levels of hunger or of breastfeeding are different from whatever trends might have been ongoing. Furthermore, it is not clear what, if anything, was done with the responses to the second questionnaire, or how these data were used.
A final remark: if health workers repeated emphasize to women that they must eat well because they are breastfeeding (and I assume that they do so), it is quite easy for a woman who cannot eat well to think that maybe she should not/cannot breastfeed.